# Alloy Design and Fabrication of Duplex Titanium-Based Alloys by Spark Plasma Sintering for Biomedical Implant Applications

**DOI:** 10.3390/ma15238562

**Published:** 2022-12-01

**Authors:** Muhammad Farzik Ijaz, Hamad F. Alharbi, Yassir A. Bahri, El-Sayed M. Sherif

**Affiliations:** 1Mechanical Engineering Department, College of Engineering, King Saud University, P.O. Box 800, Riyadh 11421, Saudi Arabia; 2Centre of Excellence for Research in Engineering Materials (CEREM), King Saud University, P.O. Box 800, Riyadh 11421, Saudi Arabia

**Keywords:** metallic implants, powder metallurgy, spark plasma sintering, microstructure, mechanical properties

## Abstract

Very often, pure Ti and (α + β) Ti-6Al-4V alloys have been used commercially for implant applications, but ensuring their chemical, mechanical, and biological biocompatibility is always a serious concern for sustaining the long-term efficacy of implants. Therefore, there has always been a great quest to explore new biomedical alloying systems that can offer substantial beneficial effects in tailoring a balance between the mechanical properties and biocompatibility of implantable medical devices. With a view to the mechanical performance, this study focused on designing a Ti-15Zr-2Ta-*x*Sn (where *x* = 4, 6, 8) alloying system with high strength and low Young’s modulus prepared by a powder metallurgy method. The experimental results showed that mechanical alloying, followed by spark plasma sintering, produced a fully consolidated (α + β) Ti-Zr-Ta-Sn-based alloy with a fine grain size and a relative density greater than 99%. Nevertheless, the shape, size, and distribution of α-phase precipitations were found to be sensitive to Sn contents. The addition of Sn also increased the α/β transus temperature of the alloy. For example, as the Sn content was increased from 4 wt.% to 8 wt.%, the β grains transformed into diverse morphological characteristics, namely, a thin-grain-boundary α phase (α_GB_), lamellar α colonies, and acicular α_s_ precipitates and very low residual porosity during subsequent cooling after the spark plasma sintering procedure, which is consistent with the relative density results. Among the prepared alloys, Ti-15Zr-2Ta-8Sn exhibited the highest hardness (s340 HV), compressive yield strength (~1056 MPa), and maximum compressive strength (~1470). The formation of intriguing precipitate–matrix interfaces (α/β) acting as dislocation barriers is proposed to be the main reason for the high strength of the Ti-15Zr-2Ta-8Sn alloy. Finally, based on mechanical and structural properties, it is envisaged that our developed alloys will be promising for indwelling implant applications.

## 1. Introduction

At present, in the field of prostheses, hip revision surgeries are expected to increase by a total of 137% between 2005 and 2030, whereas knee revision surgeries will increase by 602% in the same period. Therefore, in the pursuit of circumventing the surge of costly and painful post-implant surgeries, the life span of the indwelling implant must also be further increased. The life expectancy of the implant can be increased by enhancing its mechanical properties, such as strength, wear, and fatigue resistance [1,2,3]. 

From the viewpoint of load-bearing characteristics, metallic alloys have also been widely preferred over nonmetallic biomaterials such as polymers because, intriguingly, metallic alloys offer a higher load-bearing capacity when utilized in the manufacturing of artificial joints for hips and knees. The higher load-bearing capacity and superior wear resistance have made Ti and its alloys indispensable in implant applications [4,5].

Among traditional metallic alloys, such as stainless steel and cobalt-based alloys, α-Ti and especially (α + β)-Ti-6Al-4V alloys are an enticing class of metallic materials with higher specific strength (σ/ρ) and offer a synergistic combination of mechanical and physical properties that can recapitulate natural bone. In particular, the Ti-6Al-4V alloy is considered an α-β alloy that contains 6 wt.% aluminum as the α stabilizer and 4 wt.% vanadium as the β stabilizer. That said, using (α + β) Ti-6Al-4V-based alloys is precarious for the long-term efficacy of indwelling implants, mainly due to challenges associated with cell-mediated cytotoxicity that are mainly due to the release of toxic aluminum (Al) and vanadium (V) ions. Hence, based on biocompatibility aspects, it is highly important to develop Al- and V-free Ti-based alloys with mechanical properties closer to or higher than classical Ti-6Al-4V alloys [6,7,8].

Pure titanium (CP) occurs in a variety of allotropes. These allotropes differ widely in their crystal structure, density, and microstructural features. At the β transus temperature (approximately 885 °C), this phase is transformed into a body-centered cubic (BCC) crystal structure called the β phase. At room temperature, pure titanium has a hexagonal close-packed (HCP) crystal structure, named the α phase [9]. The α-type Ti alloy is formed by CP titanium with the addition of α-stabilizer elements such as aluminum (Al) and tin (Sn). Typically, high strength and hardness are major mechanical attributes of α-Ti alloys. Alloys that are in the β bcc phase contain significant amounts of beta stabilizer or isomorphous additions, such as molybdenum (Mo), tantalum (Ta), niobium (Nb), and vanadium (V). The fundamental mechanical characteristic of β-type Ti-based alloys is lower mechanical strength than the conventional (α + β) type of Ti-based alloys [9]. To date, through extensive compositional designs, five types of multicomponent biomedical Ti-based alloys have been discovered, which are designated as α, near-α, α + β, near-β, and β alloys [9,10,11,12,13,14,15,16,17].

The increase in strength in most metallic alloys is attributed to the precipitation strengthening phenomenon, which is controlled by factors such as the precipitate size, coherency, and strength (or hardness) of the precipitate. Precipitation strengthening occurs by two mechanisms: the shearing mechanism or the Orowan bypass mechanism, depending on many factors, including the precipitate size, coherency, antiphase boundary (APB) energy, and strength (or hardness) of the precipitate. Evidently, an important factor for enhancing the room-temperature mechanical properties of dual-phase (α + β)-Ti alloys is directly related to the phase distribution and morphological features of the α-phase structure formed at room temperature [18]. This isothermal phase decomposition is caused by a rapid rise in the free energy of the primary high-temperature BCC phase. For example, when the theoretical composition of the alloy is located near the intersection of the Gibbs free-energy curves of the α and β phases, the difference in free energy between the α and β phases provides a driving force for the direct transformation of the β to α phase in a short time during the cooling process [9,19,20]. Likewise, the physicochemical properties of Ti-based alloys can also be tailored by adjusting the beta-phase stability. In fact, the molybdenum equivalency (MoE) and beta transus temperatures (T_β_) are among the most prevalent theoretical alloy design methodologies, which are widely utilized to predict the tendency of the alpha/beta transus temperature (α/β_trans_). Whilst alloys having a compositional design between 0 ≤ MoE < 5 are beta-rich, alloys with 5 ≤ MoE < 10 are near-beta, and alloys with 10 ≤ MoE < 30 are metastable, whereas alloys with MoE > 30 are stable at room temperature [20,21]. The value of T_β_ generally decreases with the increasing MoE value. In addition, the bond order (Bo), the energy level of the metal d-orbital (Md), and the ratio of valence electrons/atom (e/a) can be exploited in other kinds of titanium-based alloys as well [22,23,24].

Likewise, to prepare biomedical alloys with various chemical components, there are two customary methods that are commonly utilized, powder metallurgy (P/M) and the conventional fusion method, also known as the ingot metallurgy method (I/M) [25]. When comparing the two preparation methods, indeed, powder metallurgy usually does not require chemical elements to melt, which facilitates the realization of components with less microstructural segregation and uniform mechanical properties [26,27]. Quite recently, additive manufacturing (AM) emerged as a novel preparation method to manufacture multicomponent alloys, but unfortunately, it is a very cost-consuming route, and secondly, it is also not suitable for refractory elements owing to the local fusion of powder particles [28].

More recently, various attempts have been made to improve the mechanical properties of titanium-based alloys by utilizing the powder metallurgical route. D. Zhao et al. [29] produced Ti-22Nb alloys by the metal injection molding method, and they found that after sintering at 1500 °C for 4 h, the MIM Ti–22Nb alloy exhibited a good combination of high yield strength (about 650 MPa) and low Young’s modulus (about 70 GPa). Similarly, E. Yilmez et al. [30,31] fabricated Ti-(25–40) Nb wt.% alloys by the powder metallurgy injection technique, and after an alloying element optimization, the hardness and rupture strength were improved when compared to titanium. In the same context, Kalita et al. [32] investigated the positive effects of Mo and Ta addition in Ti-Nb alloys prepared by mechanical alloying and spark plasma sintering. For example, the Ti-8Nb-2Mo alloy exhibited a yield strength of 806 MPa, a compressive strength of 1505 MPa, and a maximum compressive strain of 22%. Subsequently, from the literature, it is noteworthy that the hybrid powder metallurgy route coupled with high-pressure sintering is more economical because of the short production cycle, and secondly, the hot isostatic pressing (HIP) method is conducted over the α/β transus temperature (~900 °C) under high pressure, which is a very effective strategy for decreasing the porosity and residual stresses in the specimens [33,34,35]. The hardness values of the samples subjected to high-pressure sintering can also be adjusted because as the sintering temperature increases, the hardness is increased due to more refined consolidation, which is attributed to the increase in particle-to-particle bonding, high diffusion rates, and mass movement [36,37].

Despite their alluring attributes, the fabrication of Ti-based alloys by the powder metallurgical method is still moderately researched when compared to the conventional ingot metallurgy method [25,28,30,32,38]. Therefore, to produce alloys for next-generation implants, there is an urgent need to further explore the fabrication of Ti-based alloys by spark plasma sintering technology using the powder metallurgical route. Hence, the target of this work was two-fold. Firstly, on the one hand, the aim was to manipulate the powder metallurgical fabrication method for the fabrication of a fully consolidated high-strength (α + β)-type alloying system on par with traditional α-type CP Ti and (α + β) Ti-6Al-4V alloys. On the other hand, the optimal compositional design strategy was tuned to tailor a biocompatible Ti-15Zr-2Ta-*x*Sn (*x* = 4, 6, 8) alloying system. To substantiate this fact, the selection of Zr, Ta, and Sn as alloying elements was made because of their widely known superior biomechanical characteristics [38,39,40,41,42]. 

In addition, a comprehensive investigation of the effect of Sn on the relative density, microstructure, morphology phase composition, microhardness, elastic modulus, and ultimate tensile strength was also undertaken to corroborate the Sn content suitable for improving the microstructural morphology and its resulting impact on the mechanical behavior of our tailored quaternary TiZrTaSn alloying system. The experimental results indicated that adding 8 wt.% Sn to the ternary Ti-15Zr-2Ta alloy has a significant influence on the compressive strength of the prepared alloys, mainly due to the spatial distribution of the α-phase formed during the cooling stage of the sintering cycle. Among the prepared alloys, Ti-15Zr-2Ta-8Sn exhibited the highest hardness (~340 HV), compressive yield strength (~1056 MPa), and maximum compressive strength (~1470); since it does not contain any toxic elements, it is promising for biomedical applications.

## 2. Materials and Methods

### 2.1. Alloy Design and Fabrication

Theoretical methods based on beta transus temperatures (T_β_), Bo, Md, and e/a ratio parameters have been imperative tools for ensuring the stability of the β phase at room temperature for Ti-based alloys prepared by powder metallurgical (P/M) and/or ingot metallurgical routes (I/M). The theoretical chemical composition design of our four new alloying compositions with different Sn contents was used. In our theoretical design of the Ti-15Zr-2Ta-*x*Sn (*x* = 4,6,8) (all in wt.%) alloying system, the calculated values of the beta transus temperatures (T_β_) were found to be above 1000 K, whereas B_o_ ~2.80 and M_d_ ~2.48, and the average valence electron concentrations of each composition were calculated to be 4. The β transus temperature increased with the Sn content. The calculated values revealed that a Sn content of 2 wt.% in the alloy raises the α + β/β phase transition temperature by 18 °C. Hence, with this compositional design, it is foreseen that the transitional duplex refined structures of the (α + β) phase will be most plausible at room temperature when the specimens are cooled from a single β-phase region. In this work, the raw materials for the preparation of the Ti-15Zr-2Ta-*x*Sn alloying system (*x*= 4, 6, 8; hereafter, all compositions are given in wt.%.) by the powder metallurgical route were elemental powders supplied by Nanochemazone, Canada, having the following purity and average particle sizes: Ti (purity 99.9%, particle size 10 μm), Zr (purity 99.9%, particle size 5–7 μm), Ta (purity 99%, particle size 10 μm), and Sn (purity 99%, particle size 10 μm). Four specimens for each composition were prepared. A summary of the chemical compositions of the alloying elements in wt.%, along with their designations, is listed in Table 1. Hereafter, Ti-15Zr-2Ta-4Sn will be denoted as Alloy-A, Ti-15Zr-2Ta-6Sn will be denoted as Alloy-B, and Ti-15Zr-2Ta-8Sn will be denoted as Alloy-C. 

Next, the powder mixtures of each composition were subjected to mechanical milling in a planetary ball mill (Pulverisette7, FRITSCH) with a forged steel ball. A constant ball-to-powder mixture ratio was maintained during the ball-milling experiment for a horizontal rotation velocity of 150 rpm for 1hr. This step was performed to obtain a fine size and uniform distribution of the elemental powders.

To obtain reasonable consolidation, all ball-milled mixed powder specimens were subjected to high pressure (HP) in the spark plasma sintering (SPS) process. The merits of the frequency induction heat sintering furnace (HFIHS) method are that it reduces the sintering time and also consolidates the powder mixtures without the addition of a binder. Thus, our production process for powder-metallurgical-route-oriented fabrication is encompassed in the following main steps: 4 gm samples of the pure Ti, pure Zr, pure Ta, and pure Sn powder were pressed into four graphite dies. Each die had a diameter of 10 mm and a length of 200 mm. The die of each alloying composition was subjected to the high-frequency induction heat sintering (HFIHS) process. The parameters, such as sintering temperatures, times, pressures, and heating rates, were carefully chosen for the sintering of our alloying powders to achieve a superior density. After this stage, powders were loaded in electrically conducting graphite. The die was then subjected to an HFIHSF heating process at a rate of 100 K/min with a 5 min holding time and an applied load of 40 MPa. The weights of Ti and Ti-Zr powders were equal (4 gm) to obtain similar thicknesses when uniaxial pressure was applied. This was to conform to the solidified sample as per the dimensions of the die. The chamber was evacuated until it reached a vacuum of 1 × 10^−3^ Torr. After sintering, the samples were left to cool down and then removed from the die.

After the high-pressure spark plasma sintering (HFIHS) procedure, the densified specimens were ground by sandblasting to remove the remnants of graphite and/or surface inclusions formed on the surface of the specimens. Next, the specimens were subjected to density measurements. All specimens were accurately weighed in both air and deionized water using a Sartorius SECURA224-1s electronic balance with an accuracy of 0.001 mg. The density measurements were performed by following the Archimedes principle, which is commonly known as the buoyancy method. In our case, during the theoretical density calculation, the rule of mixtures was followed for each specimen while taking a theoretical density of 4.86 g/cm^3^ for the Ti-15Zr-2Ta-4Sn alloy, 4.90 g/cm^3^ for the Ti-15Zr-4Ta-6Sn alloy, and 4.95 g/cm^3^ for the Ti-15Zr-2Ta-8Sn alloy specimen [43].

### 2.2. Microstructural Characterization and Phase Constituent Analysis of the Prepared Alloys

The phase constituent analysis and microstructure observations of each sintered bulk alloy specimen after HFIHS were studied using optical microscopy (OM) and X-ray diffraction analysis (XRD). Firstly, for microstructural and phase constituent analyses, samples were taken from all four alloys across the cross-sectional surfaces of the specimens. A diamond cutting disc was mainly used to cut the sintered alloys, and the surfaces of the specimens were also made flat by a rotary disc machine. Before microstructural and phase constituent analyses, the surface of each specimen was thoroughly prepared to tweak the mirror-like surface finish. This was carried out by the following sequential steps. The specimens were first mounted in Bakelite powder (Buehler, Lake Bluff, IL, USA). Then, the specimens were ground with silicon carbide paper up to 1000 grids in water. Finally, each alloy specimen was polished using Al_2_O_3_ particles with particle sizes of 6 μm to 0.5 μm. For microstructure observations and phase constituent analysis, the following equipment and specifications were utilized. Phases of alloys were identified through X-ray diffraction (XRD) at a scanning speed of 2°/min while using an X-ray diffractometer (D-8 Discover, Bruker, Berlin, Germany) with CuK_α_ monochromatic X-ray radiation, a tube current of 15 mA, and a voltage of 30 kV. An Olympus optical microscope (Olympus BX51M, Tokyo, Japan) was used to observe the grain size and morphological analysis of the precipitated phase in the densified bulk specimens. The surfaces of the specimens were etched by using Kroll’s reagent (92 mL of water, 6 mL of nitric acid, and 2 mL of hydrofluoric acid). 

### 2.3. Mechanical Characterization of the Alloys

To determine the mechanical properties, microhardness and compression tests were performed. The hardness of the polished cross-section surfaces of specimens was measured using a Vickers micro-indenter hardness testing machine (WOLPERT UH930, Wilson Hardness, Shanghai, China). The equipment has a diamond-shaped indenter. This test was conducted with a 500 g indentation load for 10 s time. Each alloy specimen was indented at seven different positions, and average values were recorded. A compression test was carried out on the cylindrical specimens of the sintered alloys at room temperature. Cylindrical specimens had dimensions of about 6 mm in diameter and 10 mm in height. The test was conducted using a computer-controlled electronic universal testing machine (INSTRON 5984) with a cross-head speed of 0.5 mm min^−1^ and a strain rate of 2.5 × 10^−3^ s^−1^, mainly to evaluate the maximum compression strength of the specimens.

## 3. Results and Discussion

### 3.1. Evaluation of Density of the Prepared Alloys

As discussed in the previous section, the theoretical density of all specimens was calculated using the concept of the rule of mixtures, and the actual density of the specimens was calculated by Archimedes’ principle [35,43]. The values of the relative density of Alloy A, Alloy B, and Alloy C prepared by the powder metallurgical route are listed in Table 2.

From Table 2, it is inferred that due to the synergistic combination of high induction heated sintering and the Sn effect, a fully consolidated structure was achieved by the powder metallurgical route. Indeed, the HFIHS process promoted a densified solid structure with a relative density of 99% in all sintered specimens. It is worth mentioning here that the role of Sn in improving the sintering reaction should also not be neglected. This is because Sn, when compared to other elements, increases the driving force in the sintering process, mainly due to a decrease in surface energy. The driving force during the sintering process comes from the surface of the powder mixture particles. The surface energy is regarded as the governing factor in the consolidation of the powder mixture because it mitigates the neck formation between the powder particles and accelerates the solid-state diffusion process and densification rate. A similar effect of an increase in the density of the powder mixture has also been confirmed in Ti-Nb-Sn alloys [20]. 

### 3.2. Microstructural Characterization and Phase Constituent Analysis of the Prepared Alloys

Figure 1 displays the microstructural characterization of Alloy A, Alloy B, and Alloy C prepared by the powder metallurgical route. From the optical micrograph, as shown in Figure 1, the volume fraction of the α phase gradually increased with the increase in the Sn content (i.e., from Alloy-A to Alloy-C). It has been claimed that when the theoretical compositional design of alloys is located near the intersection of the free-energy curves of the α/β phases, the variation in the free energy between the two phases, that is, α and β phases, provides the driving force for the direct transformation of high-temperature β to the equilibrium α phase in a short time when compared to alloys that are designed with variables having higher beta-phase stability [20]. Thus, based on the microstructural results, it is assumed that with the addition of Sn from 4 wt.% to 8 wt.%, the conversion of the high-temperature β phase to the α phase by virtue of the spinodal phenomenon was because of more drastic furnace cooling [44,45]. Due to the high heating and cooling rates involved in the HIFHS process, grain growth did not occur in any of the sintered specimens. Meanwhile, when comparing the microstructural features of all four specimens, it can be broadly observed that all of the alloys exhibited Widmanstätten, in which the dark phase represents α, and the white phase represents β. It is noted that a very consolidated microstructure is confirmed, and a very small number of Kirkendall-type residual pores [45] are also visible in Figure 1. Nonetheless, the lack of evidence of a microstructural porosity observation in our prepared alloys validates our relative density measurements, as listed in Table 2. Interestingly, with the gradual change in Sn content, the morphology of the equilibrium phases in the alloys changed from the elongated sheet-like structure to a cross-lamellar structure to micron-scale, very fine-width pointed characteristics, as labeled in Figure 1. The major features of the α precipitate formed in all specimens are typically very diverse, and they can be subdivided into a typical thin-grain-boundary α phase (α_GB_), lamellar α colonies, and an acicular α_s_-type precipitate phase [46,47]. In our case, it is plausible that the burst increase in the precipitate could be attributed to the continuous furnace cooling after HFIHS. Since all specimens were cooled normally to room temperature, the distribution of the alpha phase is mainly dependent on our processing method [47,48]. Typically, when comparing the microstructural morphologies of Alloy-A (having 4 wt.% Sn) and Alloy-C (having 8 wt.% Sn), it is evident that Alloy-4 exhibits diverse and composite morphological characteristics ranging from a thin-rain-boundary α phase (α_GB_) and lamellar α colonies to acicular αs precipitates, as shown in Figure 1. Among the prepared alloys in this study, the number and the density of the lamellar αs phases were remarkably higher in Alloy-C. The diversity of the thin-grain-boundary α phase (α_GB_), lamellar α colonies, and acicular αs precipitates emerged owing to the solution treatment being above the β transus for a short time plus subsequent cooling after HFIHS. It was established that the elicitation of the micro-constituents in our fabricated alloys was also strongly dependent on the HFIHS parameters [48,49,50,51].

Figure 2 shows the XRD profiles of Alloy A, Alloy B, and Alloy C prepared by the powder metallurgical route. Firstly, it can be observed that no peaks corresponding to Ta, Zr, or Sn elements were indexed in any of the XRD profiles. This means that all of the specimens were densified wholly in a solid state, and all of the powder alloying elements completely diffused into the matrix during the four sequential stages of sintering in the high-frequency induction heat sintering (HFIHS) production route. This observation is consistent with the relative density results mentioned previously. Secondly, Alloy A, Alloy B, and Alloy C prepared by the powder metallurgical route specimens were all composed of α and β phases. The XRD patterns of Alloy A, Alloy B, and Alloy C prepared by the powder metallurgical route exhibited well-defined peaks at 2θ values of 35, 38, 40, and 53, which are indexed to be from the (100)α, (002)α, and (102)α planes with hcp crystal structures, and a single peak from the (110) β plane with a bb crystal structure was also indexed at a 2θ value of 40 [7,52,53]. This indicated that the addition of Sn allowed more of the α phase to form during furnace cooling after the HIFHS process. Our XRD results for all of the alloys were in good agreement with the microstructural observation results, as shown in Figure 1. It can be seen that the alloys had typical α + β phases only, and no peaks corresponding to oxides or nitrides were observed in the XRD profiles, revealing that their contents are low and will have a negligible effect on the mechanical properties of the alloys. However, in our case, it is assumed that the mechanical properties of our presently studied (α + β) Ti-15Zr-2Ta-*x*Sn (*x* = 4, 6, 8) (all in wt.%) alloying system will be very sensitive to the morphology of the precipitation phases formed after the HIFHS process [54].

### 3.3. Mechanical Characterization of the Prepared Alloys

Figure 3 displays the Vickers hardness values of Alloy-A, Alloy-B, and Alloy-C prepared by the powder metallurgical route. From the hardness results, it can be observed that when the Sn content was increased from 4 wt.% to 8 wt.%, the average hardness values increased from 315 to 342 HV. It is widely known that, in the case of biphasic microstructures, in addition to the typical solid solution strengthening phenomenon [55,56], the hardness values also strongly depend upon the abundance of harder phases within the matrix. For instance, in the case of Ti-based alloys, the β phase has less resistance to deformation when compared to the α phase. Hence, in our case, it can be concluded that, firstly, the solid solution strengthening effect attributed to the Sn content and the increase in the diverse volume fraction of the α phase [20,53,54,55,56,57] had a key influence on increasing the hardness of Alloy-C when compared to Alloy-A and Alloy-B. The favorable effect of second-phase precipitation in the matrix for improving the hardness values of Ti-based alloys is a very well-known phenomenon. In recent years, numerous studies have been conducted for tuning the process parameters of Ti alloy fabrication and also exploring new compositional designs to form favorable microstructural morphologies that can be useful in improving the mechanical properties of Ti-based alloys produced by P/M techniques. For example, Bolzoni et al. [57] discussed the relationship between the processing parameters and the properties of representative Ti-based materials. Indeed, with the adjustment of the sintering temperature and sintering time, a high value of 336 HV (Vickers hardness) was achieved in the Ti-6Al-7Nb alloy. Coherently, the reason for achieving higher hardness was partially similar to the present study; they also claimed that the increase in the hardness value is associated with a duplex (α + β) structure and the attainment of a superior relative density through the optimization of sintering parameters. Although we believe that hardness measurement is a key parameter to assess the wear performance of the materials, for the assessment of functional stability, the measurement of compressive strength is also very crucial.

The representative compressive stress and strain curve values of Alloy-A, Alloy-B, and Alloy-C prepared by the powder metallurgical route are shown in Figure 4. In addition, the specific mechanical compressive properties and their relationship with microstructural features are also summarized in Table 3. It can be observed that with the increase in the Sn content from 4 wt.% to 8 wt.%, the yield strength and compressive strength tended to increase monotonically. The yield strength and compressive strength of Alloy-A with 4 wt.% were 713 MPa and 1215 MPa. With the addition of 6 wt.% Sn, the yield strength and compressive strength of Alloy-B increased to 850 MPa and 1330 MPa, respectively. When the Sn content further increased to 8 wt.%, the yield strength and compressive strength of the alloy further increased to 1170 MPa and 1450 MPa, respectively, as seen in Figure 4. However, in parallel, the fracture strain decreases when the Sn content is increased in the Ti-15Zr-2Ta-*x*Sn (where *x* = 4, 6, 8) alloying system. Based on previous studies, it can be argued that the increases in the yield and mechanical compressive strength of our alloying system are mainly attributed to the synergistic effect of two well-known strengthening mechanisms; i.e., the solid solution strengthening effect arose from Sn addition and the precipitation strengthening mechanism [17,20,21,44,56,57]. Thus, from the current understanding and based on the microstructural observation, as shown in Figure 1, it is anticipated that the higher compressive strength of Alloy-4 is due to a large number of α/β interfaces within the matrix generated by the ultra-fine α phase precipitation [58]. This large number of interfaces successfully impeded the dislocation motion during mechanical loading at room temperature. There are several past studies corroborating that the concentration or confinement of defects, such as dislocations at the α_s_ lamellae and αs/β interface, dictates the deformation modes and/or mechanical behaviors of a dual-phase (α + β) Ti-based alloy. For example, using TEM observations, Wang et al. [59] revealed the presence of a high-dislocation-density structure in the αs lamellae and αs/β interface. On the other hand, the dislocation density in the α_p_ phase was very low due to its limited slip systems at room temperature. Due to this, they affirmed that the dislocations impeded by the α phase increased the strength of dual-phase-45,551 alloys [59,60]. On the other hand, in Table 3, it can also be noticed that the elongation to fracture and elastic modulus have quite the opposite tendency concerning mechanical strength. For instance, with the increase in the Sn content from 4 wt.% to 8 wt.%, the elongation to fracture and elastic modulus yield strength and compressive strength tended to decrease monotonically with the addition of Sn. The increase in the elastic modulus can be explained based on the microstructural analysis as well. It is well known that although the elastic modulus is an inherent property, the elastic properties of biphasic titanium alloys are related to the morphology and fractions of phases in the microstructure [8,20,44,47,53,54,56,60,61]. With the Sn addition, the proportion of the α phase was increased, so the elastic modulus was also increased, which is quite expected in our case. This is because the inherent elastic moduli of α phases with an hcp structure are higher than those of β phases with a bcc structure. It is important to mention that this value of the elastic modulus is still lower than those of commercially available Ti-6Al-4V alloys [4,6,8]. Secondly, the aim of the present study was to increase the compressive strength of a dual-phase (α + β) Ti-based alloy without affecting the elastic modulus through a compositional design. Indeed, recently, research dedicated to improving the strength-to-modulus ratio in existing implants made of alloys using a design approach has rapidly thrived for next-generation biomedical devices [60,61].

In summary, Figure 5 provides a schematic explanation of the effect of Sn on the development of key microstructural morphologies and the resulting mechanical properties of the presently studied alloying system. Firstly, if we consider the compositional effect, it can be plausible to assume that as the Sn content is increased from 4 to 6 wt.%, the formation of the α phase is also increased. As mentioned above and also seen in Figure 1, the volume fraction of the α phase increases monotonically with the increase in the Sn content, which is also highly consistent with the compositional design and fabrication process in the presently studied alloy. Consequently, the reason for the increase in mechanical properties (as shown in Table 3) during hardness testing and compression testing is in accord with our provisional theoretical design approach as well. As per our concept, we believe that the grain size in our alloys is very refined, and therefore, the mechanical behavior of our developed alloys was most likely dominated by the interface interaction mechanism rather than typical constituent-dominated deformation mechanisms, which are commonly observed in coarse-grained alloys. The formation of an intricate interface from the prior β grains upon slowing acts as a barrier for the continuous motion of lattice dislocations, which supplemented the increase in the mechanical strength of our presently studied alloys. Finally, it is noteworthy that in the current trial of a compositional design and fabrication pathway, we were able to determine that the addition of Sn to the ternary Ti-15Zr-2Ta alloy is beneficial in improving the consolidation and densification of the alloy. Simultaneously, the addition of Sn also contributed to the gradual strengthening of ternary Ti-15Zr-2Ta alloys via substitutional solid solution strengthening.

In the future, in addition to the improvement in the strength-to-modulus ratio, in vitro biological studies encompassing the response of Human Umbilical Vein Endothelial Cells (HUVECs) to the Ti-15Zr-2Ta-8Sn alloy in terms of cell viability, cell proliferation, phenotypic marker expression, and nitric oxide release will be the main scope of further investigations.

## 4. Conclusions

In the present study, we mainly aimed to develop an alloying system by using a coupling design approach, and special heed was paid to using biocompatible alloying elements, such as Ta, Zr, and Sn, and the theoretical calculation model was utilized to facilitate the design and development of the optimal processing route for the desired microstructural outcomes. Secondly, the influence of the microstructure scale and morphology on the mechanical properties of our developed system were also discussed. The following conclusions are summarized:The addition of Sn coupled with an advanced high spark plasma sintering process resulted in the production of a fully consolidated structure. The relative density of all of the prepared specimens was above 99% after the sintering process. The increase in the relative density and the overall smaller pore size of the prepared alloys could have resulted from the higher driving force that comes from the effect of the initial fine powder size of the alloying elemental powders.As expected, optical microscopic observations revealed that the addition of Sn in amounts from 4 to 8 wt.% markedly increased the volume fraction of the α phase in the matrix because Sn is considered to be an α-stabilizing element.Among all of the developed alloys, typically, Alloy-4 exhibited much higher hardness than its presently studied counterparts. Among the prepared alloys, Ti-15Zr-2Ta-8Sn exhibited the highest hardness (~340 HV). The hardness value obtained in this research study might be lower than in the literature but is still higher than bone hardness, with HV = 143.6 ± 19.Phase constituent analysis by XRD analysis revealed two phases (α + β) of the titanium alloy, which is consistent with our theoretical calculations. All of the prepared alloys have mostly alpha-phase peaks, and a β phase is also evident in XRD profiles. The formation of the α phase is related to the composition of the alloys. The α phase is formed during cooling from the temperature range of β-phase stability.The increase in the hardness of the alloys is mainly due to the solute strengthening of the α phase caused by a higher concentration of Sn.A correlation between uniaxial compression test results revealed that the underlying mechanical properties of our dual-phase alloys were sensitive to their composition and the resulting strengthening mechanism, mainly due to the morphological characteristics of the precipitated alpha phase.The precipitation of the acicular secondary α phase in the β matrix is the primary reason for the high compressive strength in our developed alloys. This is because a large number of very fine needle-shaped acicular αs precipitates can generate multiple α/β interfaces that tend to confine the dislocations under applied loading and improve the mechanical strength of the alloy.For specimens with a content of up to 8 wt.% Sn, the compressive yield strength was ~1056 MPa, and the maximum compressive strength was ~1470. This is due to the diversified distribution, as well as the composite structural morphology of the alpha phase precipitated during the cooling of the specimens after the high spark plasma sintering process.Owing to higher compressive strength and improved mechanical properties, the alloying system has good characteristics that may meet the challenges of long-term fracture in bio-implants. In particular, it lacks cytotoxic elements such as V or Al that are used in commercial medical-grade Ti-6Al-4V alloys for orthopedic and dental applications.

## Figures and Tables

**Figure 1 materials-15-08562-f001:**
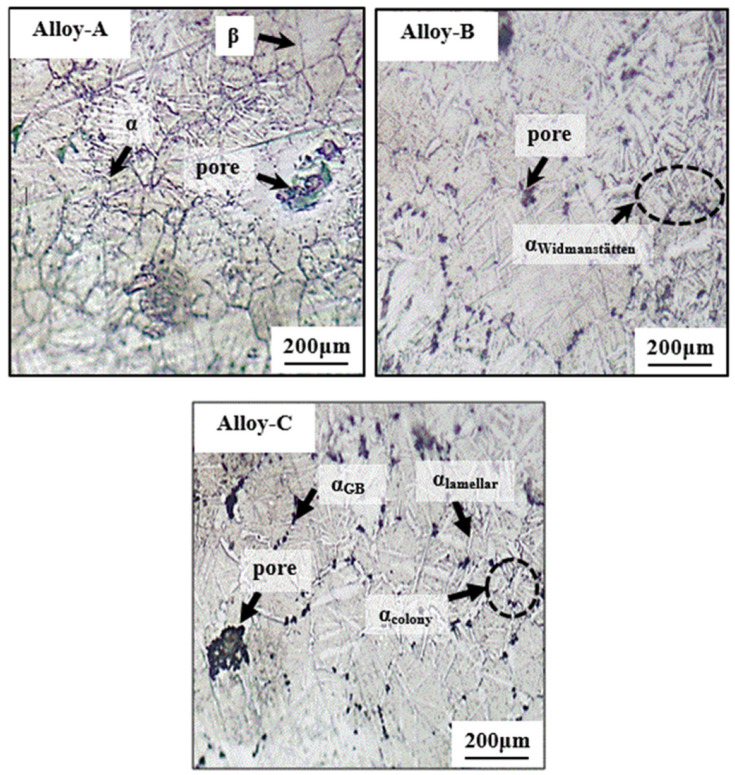
Optical micrographs of the experimental alloys fabricated by high-pressure spark plasma sintering (HFIHS) procedure.

**Figure 2 materials-15-08562-f002:**
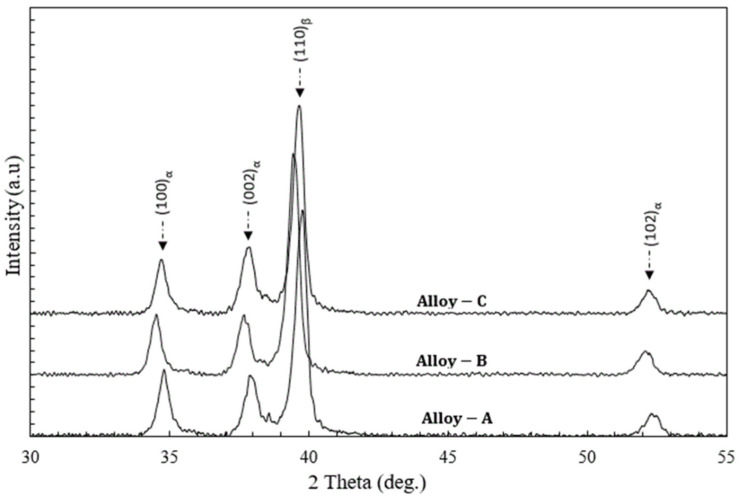
XRD patterns of the experimental alloys fabricated by high-pressure spark plasma sintering (HFIHS) procedure.

**Figure 3 materials-15-08562-f003:**
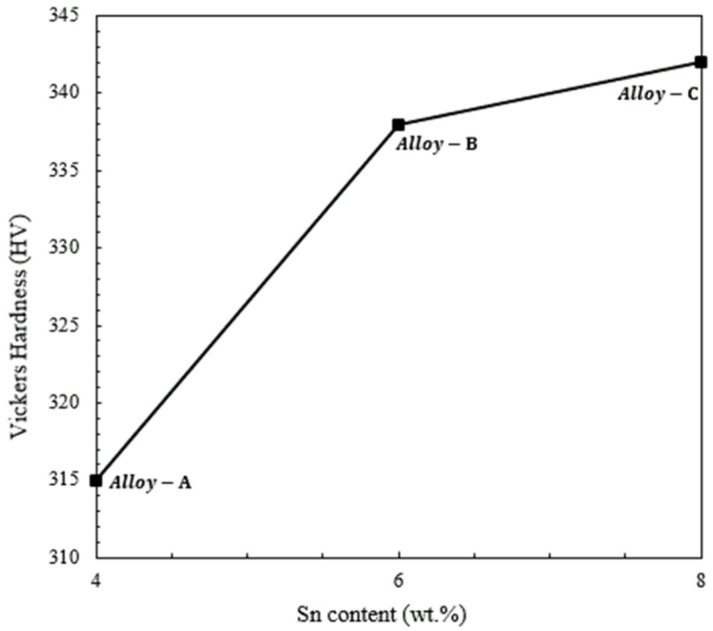
Vickers hardness values of the experimental alloys fabricated by high-pressure spark plasma sintering (HFIHS) procedure.

**Figure 4 materials-15-08562-f004:**
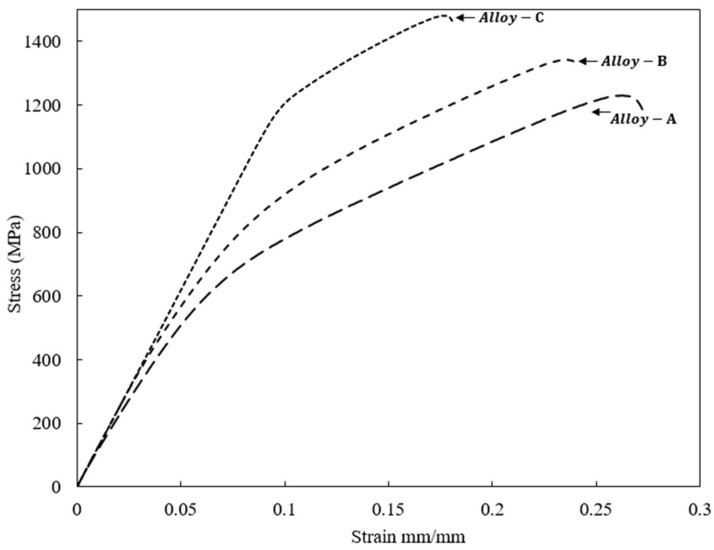
Compressive stress–strain curves of the experimental alloys fabricated by high-pressure spark plasma sintering (HFIHS) procedure.

**Figure 5 materials-15-08562-f005:**
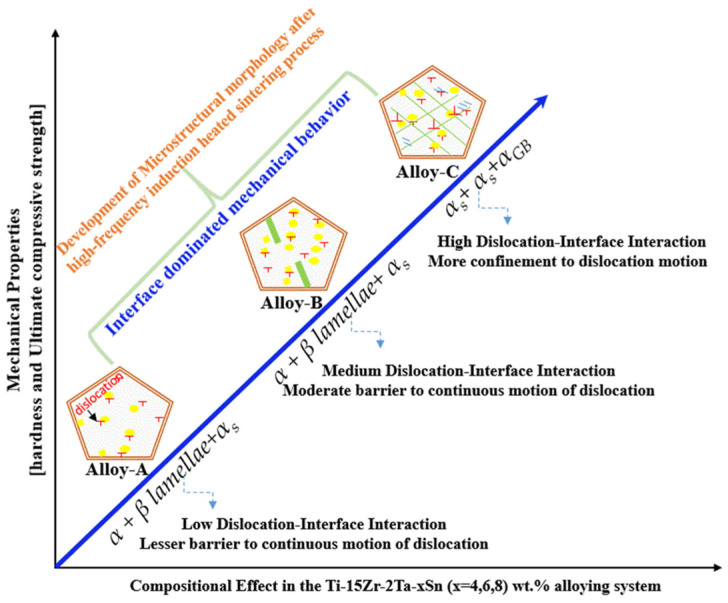
A schematic explanation showing the effect of microstructure and resulting mechanical evolution of the experimental alloys fabricated by high-pressure spark plasma sintering (HFIHS) procedure.

**Table 1 materials-15-08562-t001:** The theoretical values and chemical compositions of the designed experimental alloys.

Alloys	Transus (α/β) (K)	Valence Electron Per Atom Ratio (e/a)	Mean Bond Order (Bo)	D-Orbital Energy Level (Md)	Chemical Composition (wt.%)			
					Ti	Zr	Ta	Sn
**Alloy-A**	1160	4	2.80	2.48	79	15	2	4
**Alloy-B**	1153	4	2.80	2.48	77	15	2	6
**Alloy-C**	1135	4	2.79	2.47	75	15	2	8

**Table 2 materials-15-08562-t002:** Relative destiny of the experimental alloys fabricated by high-pressure spark plasma sintering (HFIHS) procedure.

Alloys	Theoretical Density (g/cm^3^)	Relative Density (%)
**Alloy-A**	4.86	99.46 ± 0.16
**Alloy-B**	4.90	99.55 ± 0.16
**Alloy-C**	4.95	99.80 ± 0.16

**Table 3 materials-15-08562-t003:** Summary of the mechanical and microstructural characteristics of the experimental alloys fabricated by high-pressure spark plasma sintering (HFIHS) procedure.

Alloys	Compressive Yield Stress (MPa)	Compressive Stress (MPa)	Elastic Modulus (GPa)	Fracture Strain (mm/mm)	Avg. Hardness (HV)	Phase Composition	Precipitation Phase Morphology
**Alloy-A**	713 ± 16	1215 ± 10	72 ± 16	0.22 ± 3	315	Fine-grained α + β	α + β lamellae + α_s_
**Alloy-B**	850 ± 9	1330 ± 13	80 ± 19	0.20 ± 5	338	Fine-grained α + β	α + β lamellae+ α_s_
**Alloy-C**	1170 ± 12	1450 ± 17	90 ± 22	0.18 ± 5	342	Fine-grained α + β	α_s_, + α_s_+ α_GB_

## Data Availability

The datasets generated during and/or analyzed during the current study are available from the corresponding author upon reasonable request.

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
