# Peer review of "Alloy Design and Fabrication of Duplex Titanium-Based Alloys by Spark Plasma Sintering for Biomedical Implant Applications"

_materials, 2022, doi:10.3390/ma15238562_

Round 1

Reviewer 1 Report

In this manuscript, the biocompatible Ti-based alloys were designed. The microstructure and mechanical properties of the as-prepared Ti-based alloys were studied. The following issues should be addressed before it can be accepted.

1.     The relative density of the as-prepared alloys in this work is about 99.5%. I think they should not be called porous alloys. They are just bulk alloys with pore defects. The name of “porous alloys” and the corresponding discussions in the manuscript should be deleted or modified.

2.     The tensile stress-strain curves and the tensile yield strength and elongation of the as-prepared alloys should be provided. The tensile mechanical properties are more important the compressive properties.

3.     In Figure 1, “100X” in the images should be replaced by a length scale bar.

4.     In section 3.1, “The values of the relative density of Al-loy A, Alloy B, and Alloy C prepared by the powder metallurgical route are listed in Table 1.” Here, the Table 1 in this sentence should be Table 2.

5.     The micro-CT measurements should be performed to evaluate the pores in the as-prepared alloys.

Author Response

Response to Reviewer #1 Comments

Dear Editors and Anonymous Reviewers,

The authors of Manuscript ID: materials-2043579 would like to thank the editor and appreciate Reviewer #1 for his detailed comments. We are delighted with the high-quality peer-review process. Thank you very much for your valuable suggestions and comments on our manuscript.  Those comments are of great assistance to us in improving and revising our manuscript. We have studied the comments carefully and have corrected them to the best of our knowledge in line with the suggestions made by you. We hope the manuscript after careful revisions and rebuttals meets your high standards. Please find attached a detailed response to each comment point-by-point, discussing your comments and rebuttal to the reviewer's comments to overcome these issues. Finally, we also had put heed in rephrasing the title of the article and the new title of the manuscript has been made as “Alloy design and fabrication of duplex Titanium-based alloys by spark plasma sintering for biomedical implant applications

 "Please see the attachment, the revised parts are highlighted in yellow color for your convenience of re-reviewing." 

Thank you and best regards.

Sincerely yours -On behalf of all co-authors,

Prof. Dr. Muhammad Farzik Ijaz

Dept. of Mechanical Engineering, College of Engineering, King Saud University, Saudi Arabia.

Guest Editor: Special Issue "Recent Advances in Light Alloys"

https://www.mdpi.com/journal/crystals/special_issues/J5C34O7847

email: mijaz@ksu.edu.sa  Phone: + 966114676650; Fax: + 966 114676652.

Reviewer 2 Report

Review for materials-2043579

I have reviewed the paper of “Alloy design and Mechanical Properties of Porous dual phase Titanium-based alloys for orthopedic applications” submitted for the possible publication in the Journal of Materials. Despite written in a good way, the paper has some lack of points which need to be improved. My review report is shown below:

1.      This paper is an original experimental work about the mechanical properties of some porous Ti-based alloys to use in orthopedic uses.  

2.      The aim of study must be extended at the end of the introduction. The target of the study should be specified at the end of the introduction part. Why this paper is written, what is the difference of the present submitted paper from the previous works? What is needed to for this paper? Explain all these questions at the end of the introduction part of the paper.

3.      Extend the introduction with the suggested new references.

4.      Relate the mechanical properties of three different alloys to the microstructure and explain possible reasons.

5.      Despite giving 71 references at the end of the paper, some references (refs 18, 24 and 70) are quite old (older than 1990). It is difficult to find such old references. Therefore, either delete or replace them with the recent references. Several recent papers are suggested to be cited below:

a.      Eren Yılmaz, Azim Gökçe, Fehim Findik, Ozkan Gulsoy. “Assessment of Ti–16Nb–xZr alloys produced via PIM for implant applications”, Journal of Thermal, Analysis and Calorimetry, Vol: 134, issue: 1, pp. 7-14, Oct. 2018.

b.      Eren Yılmaz, Azim Gökçe, Fehim Findik, Ozkan Gulsoy. “Metallurgical properties and biomimetic HA deposition performance of Ti-Nb PIM alloys”, Journal of Alloys and Compounds, Vol. 746, pp. 301-313, May 2018.

6.      The authors should discuss the results and compare the results with the previous studies and mention coherent/incoherent points with the possible reasons.

7.      Finally, I believe the submitted paper can only be accepted after the correcting and/or adding the required points above mentioned for the publication in the journal. I also believe that this paper might be beneficial for the academicians who are working in the specific area.

Author Response

Response to Reviewer #2 Comments

Dear Editors and Anonymous Reviewers,

The authors of Manuscript ID: materials-2043579 would like to thank the editor and appreciate Reviewer #2 for his detailed comments. We are delighted with the high-quality peer-review process. Thank you very much for your valuable suggestions and comments on our manuscript. Those comments are of great assistance to us in improving and revising our manuscript. We have studied the comments carefully and have corrected them to the best of our knowledge in line with the suggestions made by you. We hope the manuscript after careful revisions and rebuttals meets your high standards. Please find below the enclosed file revealing a detailed response to each comment point-by-point, discussing your comments and rebuttal to the reviewer's comments to overcome these issues. Finally, we also had put heed in rephrasing the title of the article and the new title of the manuscript has been made as Alloy design and fabrication of duplex Titanium-based alloys by spark plasma sintering for biomedical implant applications  "Please see the attachment, the revised parts are highlighted in yellow color for your convenience of re-reviewing." 

Thank you and best regards.

Sincerely yours -On behalf of all co-authors,

Prof. Dr. Muhammad Farzik Ijaz

Dept. of Mechanical Engineering, College of Engineering, King Saud University, Saudi Arabia.

Guest Editor: Special Issue "Recent Advances in Light Alloys"

https://www.mdpi.com/journal/crystals/special_issues/J5C34O7847

email: mijaz@ksu.edu.sa  Phone: + 966114676650; Fax: + 966 114676652.

Reviewer 3 Report

The paper is devoted to the study of the mechanical properties of titanium alloys for orthopedic purposes. The topic is topical and popular among researchers. However, when reviewing the paper, I had several questions and comments. I hope my comments will help the authors improve their work and publish it in a highly rated scientific journal.

1. The title of the article contains the term Porous dual phase Titanium-based alloys. However, the article examines material with a Relative density of 99.46...99.80%. In my opinion, the term porous material is inappropriate here.

2. To estimate the number and size of pores, it is better to use a microsection without etching. As an example, you can see the work DOI 10.1088/1757-899X/248/1/012012

3. In Figure 3, there are no numerical values ​​on the x-axis indicating the concentration of Sn.

4. In the conclusions of paragraph 4, it is said that "Among the prepared alloys the Ti-15Zr-2Ta-8Sn exhibited the highest hardness (~ 340 HV) which is also larger than the commercial Ti-6Al-4V alloy having a hardness value of 183 VHN". The statement about the hardness of alloy 183 HV is not true. The hardness of Ti-6Al-4V alloy can reach 500 HV (https://doi.org/10.3390/ma12193269) and 511 (https://doi.org/10.3390/machines8040079). This conclusion needs to be clarified.

5. What is the rationale for choosing a method for obtaining blanks for orthopedic applications high-pressure spark plasma sintering?

6. The title of the article does not reflect the essence of the study. The name needs to be changed.

7. Research findings must be compared with the purpose of the work. No porous material was obtained in this work.

Author Response

Response to Reviewer #3 Comments

Dear Editors and Anonymous Reviewers,

The authors of Manuscript ID: materials-2043579 would like to thank the editor and appreciate Reviewer #3 for his detailed comments. We are highly satisfied with the high-quality peer-review process. Thank you very much for your valuable suggestions and comments on our manuscript. Those comments are of great assistance to us in improving and revising our manuscript. We have studied the comments carefully and have corrected them to the best of our knowledge in line with the suggestions made by you. We hope the manuscript after careful revisions and rebuttals meets your high standards. Please find enclosed a detailed response to each comment point-by-point, discussing your comments and rebuttal to the reviewer's comments to overcome these issues. Finally, we also had put heed in rephrasing the title of the article and the new title of the manuscript has been made as Alloy design and preparation of duplex Titanium-based alloys by spark plasma sintering for biomedical implant applications

Please see the attachment, the revised parts are highlighted in yellow color for your convenience of re-reviewing.

Thank you and best regards.

Sincerely yours -On behalf of all co-authors,

Prof. Dr. Muhammad Farzik Ijaz

Dept. of Mechanical Engineering, College of Engineering, King Saud University, Saudi Arabia.

Guest Editor: Special Issue "Recent Advances in Light Alloys"

https://www.mdpi.com/journal/crystals/special_issues/J5C34O7847

email: mijaz@ksu.edu.sa  Phone: + 966114676650; Fax: + 966 114676652.

Round 2

Reviewer 1 Report

Most of the questions have been addressed by the authors. I recommend to accept it for publication.

Reviewer 3 Report

The authors answered my questions and corrected the paper in accordance with my comments. I recommend the article for publication in this form.